# Induction of fetal hemoglobin: Lentiviral shRNA knockdown of *HBS1L* in β⁰-thalassemia/HbE erythroid cells

**Sukanya Chumchuen** [1]☯, **Orapan Sripichai** [2]☯¤, **Natee Jearawiriyapaisarn** [2], **Suthat Fucharoen** [2], **Chayanon Peerapittayamongkol** [1]*

1 Department of Biochemistry, Faculty of Medicine Siriraj Hospital, Mahidol University, Bangkok, Thailand,
2 Thalassemia Research Center, Institute of Molecular Biosciences, Mahidol University, Nakhon Pathom, Thailand

☯ These authors contributed equally to this work.
¤ Current address: Department of Medical Sciences, National Institute of Health, Ministry of Public Health, Nonthaburi, Thailand
* chayanon.pee@mahidol.ac.th, chayanon@msn.com

**Data Availability Statement:** All relevant data are within the paper and its Supporting information files.

## Abstract

Imbalanced globin chain output contributes to thalassemia pathophysiology. Hence, induction of fetal hemoglobin in β-thalassemia and other β-hemoglobinopathies are of continuing interest for therapeutic approaches. Genome-wide association studies have identified three common genetic loci: namely β-globin (*HBB*), an intergenic region between *MYB* and *HBS1L*, and *BCL11A* underlying quantitative fetal hemoglobin production. Here, we report that knockdown of *HBS1L* (all known variants) using shRNA in early erythroblast obtained from β⁰-thalassemia/HbE patients triggers an upregulation of γ-globin mRNA 1.69 folds. There is modest perturbation of red cell differentiation assessed by flow cytometry and morphology studies. The levels of α- and β-globin mRNAs are relatively unaltered. Knockdown of *HBS1L* also increases the percentage of fetal hemoglobin around 16.7 folds when compared to non-targeting shRNA. Targeting *HBS1L* is attractive because of the potent induction of fetal hemoglobin and the modest effect on cell differentiation.

## Introduction

The most common hereditable anemia in Thailand is thalassemia. There are two major types of thalassemia, α- and β-. The disease forms of β-thalassemia in Southeast Asia occur by a combination of β⁰ with β⁺, and the most common form of β⁺ is hemoglobin E [1]. G to A substitution at codon 26 of β-globin transcript leads to glutamate replaced by lysine residues. This mutation also generates an abnormal splice site resulting in an aberrant RNA processing competing with the normal splice site of βᴱ-globin mRNA culminating in a reduction in βᴱ-globin mRNA production and makes hemoglobin E a mild form of β⁺-thalassemia [2].

The symptoms of β-thalassemia/HbE display a complex trait [3, 4]. There are numerous modifiers, for example, coinheritance with α-thalassemia [5], the number of α-globin genes, types of β-globin mutations [6], and modifiers of HbF levels [7]. The gold standard treatment

**Funding:** Yes. This work was supported by Siriraj Research Fund and National Research Council of Thailand (NRCT). Funders play no role in the study design, data collection and analysis, decision to publish, or preparation of the manuscript.

**Competing interests:** The authors have declared that no competing interests exist.

for β-thalassemia is an allogenic bone marrow transplantation. With several constraints such as applicable to the relatively young age patients and suitable HLA-matched donors make this curative approach unavailable for most β-thalassemia patients [8]. Alternatively, gene-therapy approaches to elicit the induction of β- or γ-globins, can diminish the load of unpaired α-globin chains. This will reclaim the balance of α/β-like globin ratios and thereby amend the ineffective erythropoiesis and shorten RBC lifespan [9]. Transduction of stem cells with lenti-virus carrying short hairpin sequence (shRNA) targets to *BCL11A* and *SOX6* has been shown to effectively induce γ-globin mRNA expression [10, 11].

A genome-wide association study (GWAS) from a European twin cohort with significantly different F-cell values revealed that 10% of the variability is attributable to the *HBB* region, 19% in *HBS1L-MYB* intergenic region (HMIR), and 15% in *BCL11A* on chromosome 2 [12]. GWAS study in Thai β$^0$-thalassemia/HbE patients exhibits an association of the same three regions with the disease severity and HbF levels [13].

In humans, *HBS1L* is identified as one of 5 protein-coding genes (*AHI1*, *MYB*, *ALDH8A1*, *HBS1L*, *and PDE7B*) in around 1.5 Mb region associated with HbF levels on the chromosome 6q23 [14]. HBS1L functions in combination with PELOTA in identifying the stalled ribosome complexes on truncated mRNAs and causing them to be separated into subunits [15]. Additionally, this incident starts mRNA degradation pathways.

We previously indicated that an SNP in exon 1 of *HBS1L* was associated with fetal hemoglobin and disease severity in Thai Chinese β$^0$-thalassemia/HbE patients [16]. Expression of *HBS1L* is significantly diminished in erythroblasts throughout Phase II culture obtained from individuals with hereditary persistent fetal hemoglobin (HPFH) [17]. Here we extended that suppression of *HBS1L* expression by shRNA triggered considerable induction of γ-globin expression in both mRNA and protein levels while did not affect both α- and β-globin expression. Moreover, unlike any HbF inducer known so far, HbF's induction slightly perturbed erythroid differentiation characterized by flow cytometry.

## Materials and methods

### Subjects

This study was permitted by the Ethical Committee, Siriraj Institutional Review Board (Si 290/2015). All participants aged greater than 18 years old. The 5 β$^0$-thalassemia/HbE patients were recruited from the Thalassemia clinic, Nakhon Pathom Hospital, Thailand, and the 5 healthy volunteers were recruited at the Institute of Molecular Bioscience, Mahidol University, Thailand (S1 Table). The β$^0$-thalassemia/HbE patients did not receive blood transfusion for at least 1 month before participation. The healthy donors did not carry any known types of thalassemia or hemoglobin variants. All participants provided their written informed consent to be a part of the study.

### Sample collection and CD34$^+$ cell differentiation

Mononuclear cells (PBMCs) were isolated from the peripheral blood by gradient centrifugation. First, plasma was removed from the peripheral blood samples after centrifugation at 600g for 10 min at room temperature. The packed cells were then diluted with 1X Dulbecco's phosphate-buffered saline (DPBS) containing 2 mM EDTA. The diluted cell solution at the volume of 25 mL was layered on 15 mL of lymphoprep™ density medium (density 1.077 g/mL) (Axis-shield PoC AS, Oslo, Norway), followed by centrifugation at 600g for 20 min at room temperature without break. The PBMCs layer located between plasma layer and lymphoprep™ layer was carefully harvested and washed twice with the 1X DPBS buffer containing 2 mM EDTA and once with 1X DPBS containing 2 mM EDTA and 0.5% fetal bovine srum (FBS). Primary

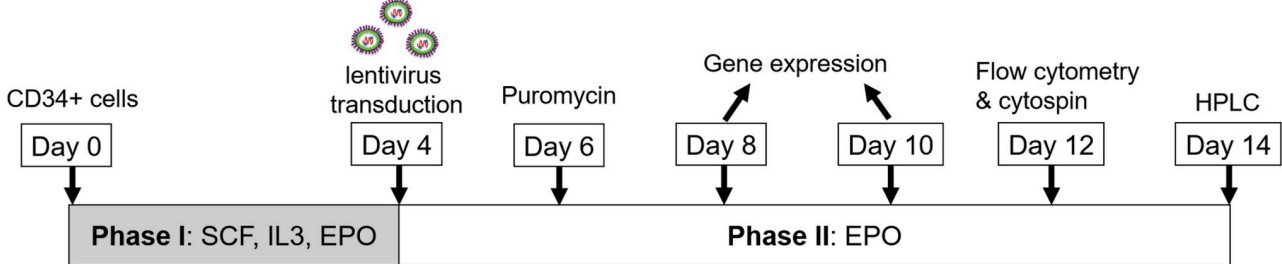

**Fig 1. A methodological scheme illustrates the culture of CD34$^+$ cells, lentiviral transduction conditions, erythroid differentiation protocols, and timeframes for gene expression, morphology, and HPLC studies.**

CD34$^+$ hematopoietic progenitor cells were positively selected from the PBMCs using MACS$^®$ CD34 MicroBead kit (Miltenyi Biotec GmbH, Bergisch Gladbach, Germany) according to manufacturer's protocol.

## Erythroid cell culture condition

CD34$^+$ cells were individually cultured in a 2-phase medium systems to drive the cellular commitment to the erythroid lineage and differentiate into mature red blood cells. Cells from each donor were cultured in Phase I medium: Iscove's Modified Dulbecco's Medium (IMDM, Gibco$^®$, Thermo Fisher Scientific, Inc., MA, USA) containing 20% FBS (Sigma-Aldrich$^®$, Sigma-Aldrich, Inc., MO, USA), 300 μg/mL holo-Transferrin (holo-TF, PromoCell$^®$, Promo-Cell GmbH, Heidenberg, Germany), 50 ng/mL human Stem Cell Factor (hSCF, CellSignaling Technology$^®$, Cell Signaling, Inc., MA, USA), 10 ng/mL interleukine-3 (IL-3, CellSignaling Technology$^®$), 2 U/mL erythropoietin (EPO, EPREX$^®$, Janssen-Cilag, Auckland, NZ) in the presence of 100 U penicillin/streptomycin (Gibco$^®$). On day 5, cells were replaced with fresh Phase II medium: IMDM containing 20% FBS, 300 μg/mL holo-TF, 5 U/mL EPO and 100 U penicillin/streptomycin at 37˚C under 5% $CO_2$ and 100% humidity. Fig 1 displays a flowchart of experimental design for cell culture, lentiviral transduction settings, and strategies for erythroid differentiation.

## ShRNA-carrying lentivirus vector development, selection, and transduction

For knockdown tests, lentivirus with *HBS1L* shRNAs was created. *HBS1L* shRNA1 (TRCN0000353597) and *HBS1L* shRNA2 (TRCN0000353653), two distinct target sense sequences of *HBS1L* shRNA, were chosen for this study because they were perfectly matched to *HBS1L* transcripts. While shRNA2 targeted all variants of *HBS1L*, shRNA1 only targeted the spliced variant transcripts 1 and 2. Table 1 lists the target sequences for shRNA1, shRNA2, and shNTC. A diagram of the *HBS1L* variants shows the shRNA-target locations (shRNA1 and shRNA2) (S1 Fig).

**Table 1. The table demonstrates the target sequence of *HBS1L* shRNAs and non-targeted shRNA control.**

| shRNA name | Target sense sequence (5' to 3') |
|---|---|
| *HBS1L* shRNA1 (TRCN0000353597) | GCGATCTATTGACAAACCTTT |
| *HBS1L* shRNA2 (TRCN0000353653) | CGTCTTTATTCATGCCTTGAT |
| non-targeted shRNA (SHC016V) | GCGCGATAGCGCTAATAATTT |

To transform bacteria, *HBS1L* shRNAs were ligated into the pLL-Puro, a modified version of the pLL3.7 lentiviral plasmid. In this plasmid, the EGFP gene was replaced by a puromycin-resistant gene [18]. By digesting them with the *Xho*I and *Xba*I restriction enzymes, five clones of each of the *HBS1L* shRNA1 and *HBS1L* shRNA2 were chosen and characterized for insertion. Direct sequencing was used to verify the insert sequences.

The lentiviral vector plasmids previously chosen were co-transfected into HEK293T cells along with the three packaging plasmids pMDLg/pRRE, pRSV-Rev, and pMD2.G using the X-tremeGENE HP transfection reagent (Roche Mannheim Germany). Puromycin (Invitrogen, Carlsbad, CA, USA) selection was used to titrate lentiviral particles from the cultured supernatant at 48 and 72 hours after transfection in order to calculate the infection multiplicity (MOI). On day 4 of erythroblast culture, lentivirus containing shRNA was transduced with a MOI of 20 in 500 μL of Phase II media supplemented with 8 μg/mL of polybrene (Sigma-Aldrich®) for 24 hours before being further cultivated in new Phase II medium for an additional 24 hours. Cells were then under selection in the presence of 1 μg/mL puromycin for 48 h, replaced with fresh Phase II medium without puromycin and continued culture until day 14.

At day 8 of culture, untransduced cells and normal erythroblast cells transduced with *HBS1L* shRNA1 and shRNA2, and shNTC (non-targeted shRNA) were extracted for RNA and protein extractions. According to the results of qPCR and Western blot, the knockdown of *HBS1L* by shRNA2 was more effective than that by shRNA1.

## RNA isolation, reverse transcription, and quantitative polymerase chain reaction (qPCR)

Total RNAs were isolated from cultured erythroblast cells on days 6, 8 10, 12, and 14 using TRIzol reagent (Invitrogen) and converted intocomplementary DNAs (cDNAs) using Super-Script III reverse transcriptase with oligo-dT primer (Invitrogen).

The synthesized cDNAs were quantified with specific primers for *HBS1L* transcripts using SYBR master mix (Applied Biosystems) according to the manufacturer's recommended conditions. Expression of α-, β-, and γ-globin was measured by SYBR green-based qPCR using primer sequences [19]. Quantitative PCR was performed on CFX96™ Real-Time system (Bio-Rad). The expression of α-, β-, and γ-globin mRNA in shNTC and shHBS1L transduced cells were calculated by $2^{-\Delta\Delta Ct}$ methods relative to untransduced (UNT) control as described below.

$$\Delta Ct_{UNT,shNTC,shHBS1L} = Ct_{GeneX} - Ct_{ACTB} \tag{1}$$

$$\Delta\Delta Ct_{shNTC,shHBS1L} = \Delta Ct_{shNTC,shHBS1L} - \Delta Ct_{UNT} \tag{2}$$

In comparison to untransduced cells, the abundance of the mRNAs for the following erythroid-related transcription factors, namely *BCL11A*, *ZBTB7A*, *KLF1*, *GATA1*, *GATA2*, *MYB*, and *ATF4*, was measured and displayed as a fold change [20]. The primer sequences utilized in this study are listed in S2 Table.

## Western blot analysis

According to the manufacturer's protocol, approximately $1X10^6$ erythroid cells were collected for protein extraction by NE-PER (Nuclear and Cytoplasmic Extraction Kit (Thermo Fisher Scientific, Inc, MA, USA). Protein concentrations were measured by Quick Start Bradford Protein Assay (Bio-Rad Laboratories, Inc.). The extracts were electrophoresed onto 12% SDS-polyacrylamide gel, transferred to polyvinylidene difluoride (PVDF) membrane, and blocked with 5% skim milk. The membranes were reacted with HBS1L specific primary antibody (NBP1-85123; Novus Biologicals, CO, USA), followed by goat anti-rabbit secondary antibody

conjugated with horseradish peroxidase (ab97051; Abcam, Cambridge, UK). Enhanced chemiluminescence (ECL; Amersham GE Healthcare, Little Chalfont, UK) was utilized as a substrate for protein visualization by Azure™ c400 Imaging System (Azure Biosystems, Inc., CA, USA). Anti-actin (ab49900) 1:50,000 (Abcam®, Cambridge, UK) and anti-lamin A (L1293) 1:5000 (Sigma-Aldrich, Inc., MO, USA) were utilized as cytoplasmic and nuclear loading controls, respectively.

### Flow cytometry analysis

The cultured erythroid cells ($5X10^4$) were collected and stained in 100 μL of 1X DPBS containing phycoerythrin (PE)-conjugated mouse monoclonal anti-human CD71 (transferrin receptor; BioLegend®, CA, USA) and allophycocyanin (APC)-conjugated mouse monoclonal anti-human glycophorin A (CD235a; BD Pharmingen™, BD Biosciences, NJ, USA). After incubation for 15 minutes at room temperature in the dark, the stained cells were diluted with 100 μL of 1X DPBS, and data were acquired by a flow cytometer (BD FACSCalibur™, BD Biosciences, NJ, USA). The FlowJo V10 (FlowJo LLC, Ashland, OR, USA) was applied for result analysis.

### Wright-Giemsa staining

Cell morphology was evaluated from $5X10^4$ of erythroid cells. The cells were spun by cytospin (StatSpin CytoFuge 2, Beckman Culter, Inc., CA, USA), stained with modified Wright-Giemsa dye (Sigma-Aldrich, Inc., MO, USA) for 5 minutes, and washed twice with water for two minutes. The slides were dried and cleaned with 70% ethanol before cell morphology observation under an Olympus CX31 light microscope.

### Hemoglobin measurement by high-performance liquid chromatography (HPLC)

The remaining erythroid cells on day 14 were all collected and washed once with 1X DPBS. The cell pellet was lyzed for hemoglobin measurement by HPLC using the VARIANT™ II β-thalassemia Short Program (Biorad, Hercules, CA, USA). This cation exchange HPLC technique uses an increasing sodium phosphate gradient buffer to separate different hemoglobin variants. Hemoglobin was measured using absorbance at 415 nm, and correction for turbidity was done by the absorbance at 690 nm [21]. Each 6.5-minute assay detects the most prevalent aberrant hemoglobin variations while also providing quantitative findings for the percentages of HbA2/E and HbF.

### Statistical analysis

The analysis was conducted in R version 4.03 [22]. The Shapiro-Wilk test was performed to check for normal distribution. Differences between suitable groups were checked with paired t-test. We used one-tail t-test for comparison of globin mRNAs and two-tail t-test for the rest of comparisons. The significant levels of less than 0.05 were employed.

## Results

### *HBS1L* expression during erythroid differentiation

*HBS1L* expressed at a low level but detectable in the early to intermediate stage of erythroid differentiation (day 6–10) and declined in the late stage (day12-14). This similar expression was observed in both healthy donors and $β^0$-thalassemia/HbE patients in Fig 2. There were none that attained statistical significance at any time.

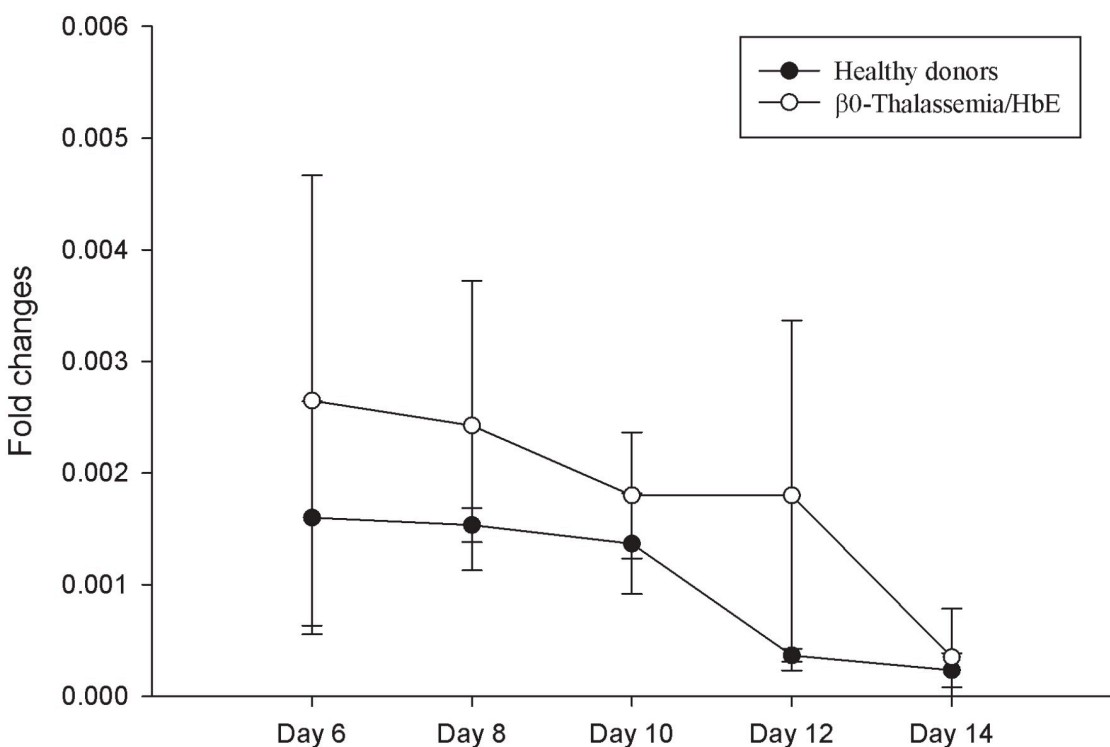

**Fig 2. Relative expression of *HBS1L* during erythroid differentiation in healthy donors (N = 3) and β⁰-thalassemia/HbE patients (N = 4).** Data is presented as mean±SD.

## Evaluation of *HBS1L* shRNA effectiveness

Erythroblasts on day 4 were transduced with lentivirus carrying *HBS1L* shRNA1, shRNA2, and shNTC. Healthy cells were collected at day 8 for RNA and protein studies. Fig 3A, the expression of *HBS1L* mRNA reduced in cells transduced with *HBS1L* shRNA2 down to 0.25 (p-value = 0.0004, N = 3) compared with *HBS1L* shRNA1 down to 0.87. These results also corresponded with the reduction of HBS1L protein localized to both cytoplasmic and nuclear portions (Fig 3B). In our hands, we could not separate isoforms of HBS1L in SDS-PAGE. Accordingly, we managed to transduce erythroblasts with *HBS1L* shRNA2 or shNTC on day 4 for the subsequent experiments.

## Effects of *HBS1L* knockdown on erythroid differentiation

Erythroblasts were transduced with *HBS1L* shRNA2 or shNTC on day 4. Cells were harvested on day 12 for differentiation studies as cells undergoing differentiation to R1 (CD71^high^CD235a^minus^), R2 (CD71^high^CD235a^high^), R3 (CD71^medium^CD235a^high^) and R4 (CD71^low^CD235a^high^) populations [23, 24] (Fig 4A and 4B). In case of β⁰-thalassemia/HbE, the mean percentages and SEM of cell populations were 0.65±0.14 for R1, 65.50±3.00 for R2, 30.63±3.93 for R3 and 0.10±0.05 for R4 in untreated erythroblasts, 0.60±0.14 for R1, 72.47±1.76 for R2, 24.73±1.20 for R3 and 0.05±0.03 for R4 in shNTC-transduced erythroblasts and 1.06±0.23 for R1, 81.83±2.72 for R2, 13.50±3.00 for R3 and 0.07±0.03 for R4 in shRNA2-transduced erythroblasts (Fig 4B and 4D). Among healthy donors, the mean percentages and SEM of cell populations were 1.92±0.17 for R1, 31.60±2.40 for R2, 61.65±2.85 for R3 and 1.47±0.18 for R4 in

untreated erythroblasts, 1.12±0.08 for R1, 24.30±0.06 for R2, 70.30±0.20 for R3 and 0.74±0.15

A

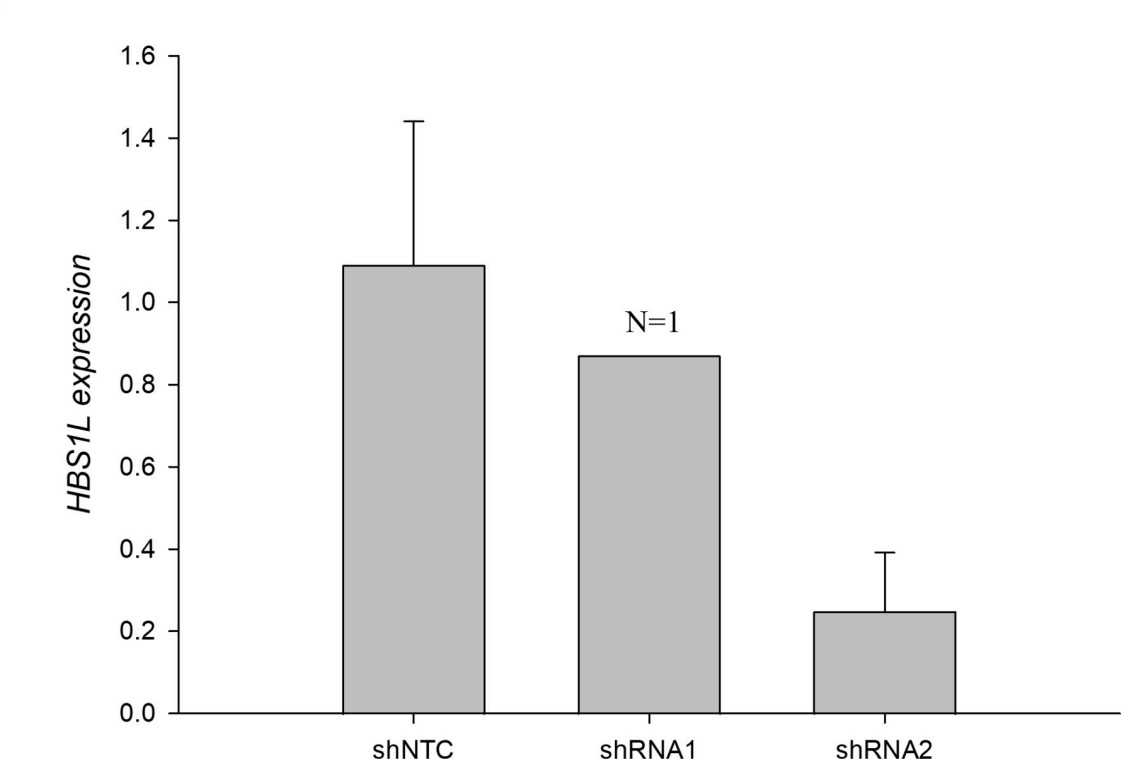

B

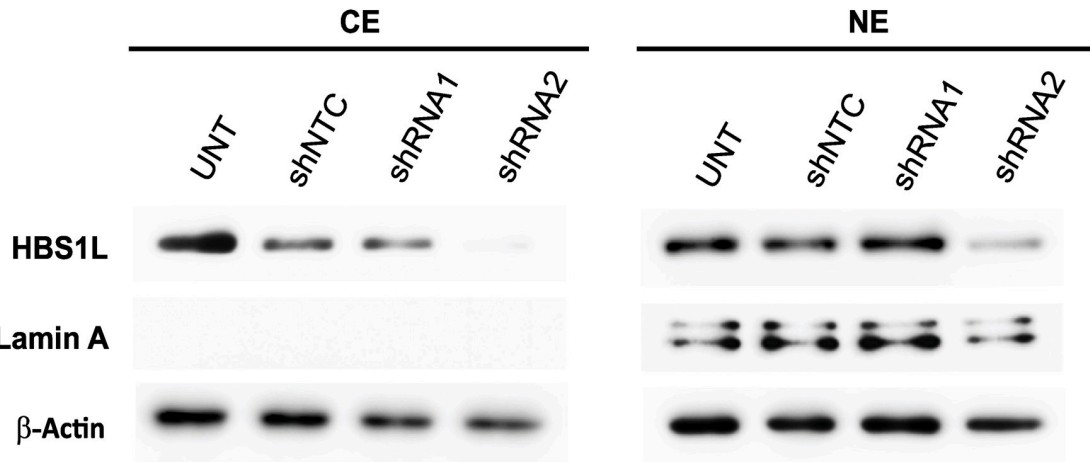

**Fig 3. Comparison of *HBS1L* shRNA effectiveness.** The scale bars represent Means ±SD. Effectiveness of *HBS1L* knockdown by shRNAs (A). Western blot was performed on cultured day 8 healthy erythroblasts transduced with *HBS1L* shRNA1, shRNA2 and shNTC and untransduced cells (UNT) (B). CE represented cytoplasmic extraction and NE for nuclear extraction, using β-actin and lamin A as a loading control.

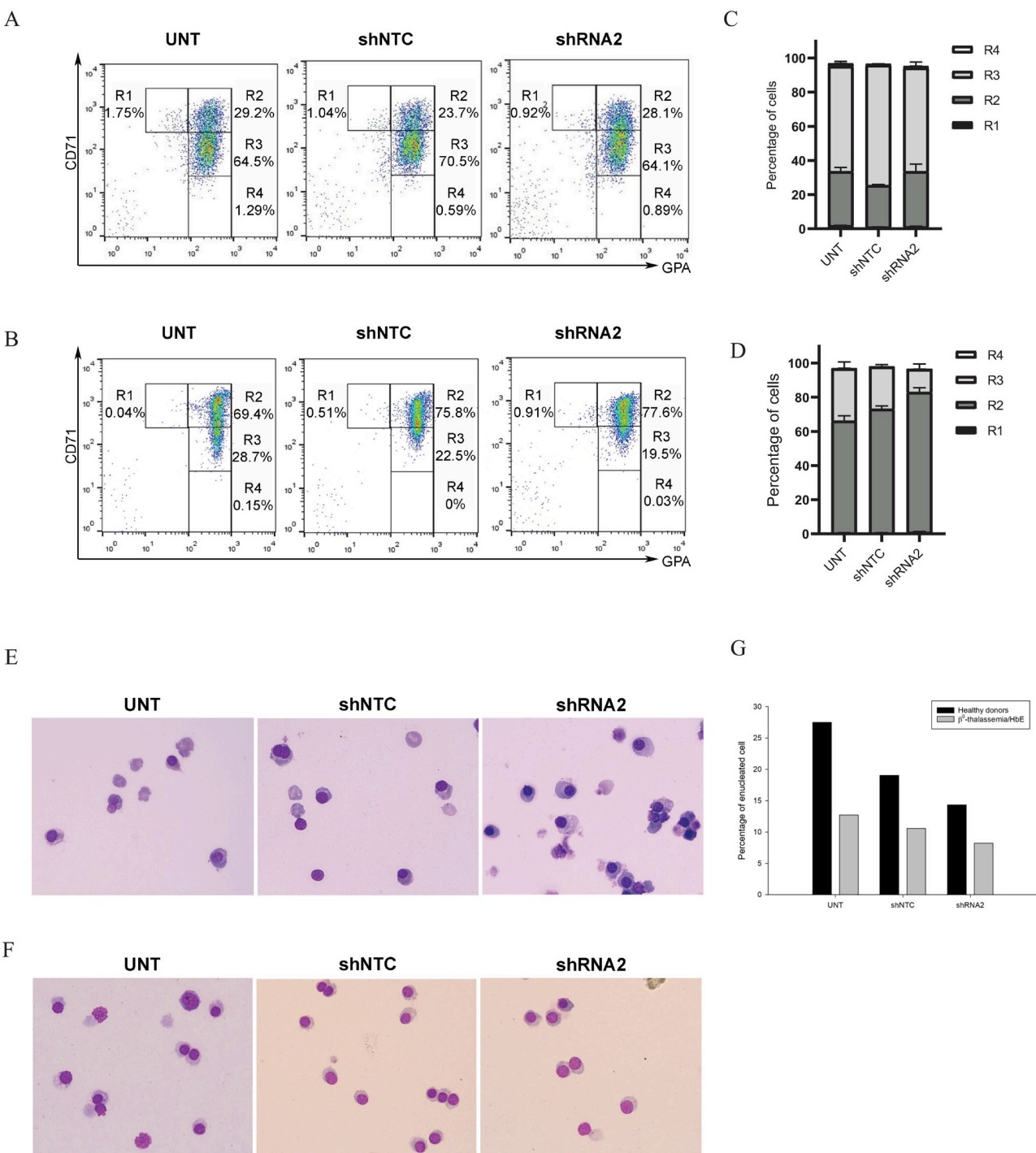

**Fig 4. Flow cytometry analysis of erythroblasts on day 12 in different conditions.** Representative picture of gated cell populations from early- to late-stage erythroblast (R1 to R4) with different levels of CD71 and GPA in healthy donors (A) and $\beta^0$-thalassemia/HbE patients (B) were presented. Mean ± SEM of each cell population was displayed in the bar graph generating from 2 healthy donors (C) and 3 $\beta^0$-thalassemia/HbE patients (D). Cell morphology of erythroid cells on day 12 stained with Wright-Giemsa dye: healthy donor (E); $\beta^0$-thalassemia/HbE patient (F). The mean percentages of enucleated cells were shown as bar graph, data was collected from 2 healthy donors and 2 $\beta^0$-thalassemia/HbE patients (G).

for R4 in shNTC-transduced erythroblasts and 1.20±0.28 for R1, 32.40±4.30 for R2, 60.70 ±3.40 for R3 and 0.86±0.03 for R4 in shRNA2-transduced erythroblasts (Fig 4A and 4C).

The flow cytometry revealed a slight increase in the percentages of R2 populations upon either shRNA2 or shNTC knockdown in $\beta^0$-thalassemia/HbE patients (Fig 4D, N = 3). The R2 populations reflect the basophilic erythroblasts. This increment indicated a slight delay of erythroid cell differentiation after *HBS1L* knockdown in $\beta^0$-thalassemia/HbE erythroblasts compared with untransduced but not shNTC transduced cells. It was noteworthy that regardless of treatment, cells were morphologically indistinguishable in Fig 4E and 4F. Fig 4G displayed the percentage of enucleation for each group. However, in the cases of shNTC and shRNA2 knockdown, the percentage of enucleated cells tended to decrease (N = 2, each). In the cells of the patients, the effects of *HBS1L* knockdown on enucleation appeared to be less.

## Effect of *HBS1L* knockdown on hemoglobin production

To evaluate the impact of *HBS1L* knockdown on α-, β- and γ-globin expression, erythroblasts were harvested for RNA extraction on day 8 in healthy donors and day 10 in $\beta^0$-thalassemia/ HbE patients, due to the low number of cells. The results indicated that *HBS1L* knockdown effectively induced γ-globin gene expression up to 1.69-fold at mRNA level in patients (compared with shNTC, p-value = 0.0117) and 1.44 folds in healthy donors (compared with shNTC, p-value = 0.05048), but there was no significant effect on α- or β-globin mRNA levels (Fig 5A and 5B). On average, we found a reduction of *HBS1L* mRNA expression of 71% in $\beta^0$-thalassemia/HbE patients and 84% in healthy donors.

HPLC analysis of hemoglobin types revealed markedly elevated fetal hemoglobin levels in all cases after transduction. Fetal hemoglobin was indeed elevated regardless of baseline levels (untransduced). The base line fetal hemoglobin levels were inferred by the untransduced conditions (healthy donors and patients). The fetal hemoglobin was increased in additional 16.7% (p-value = 0.0085) in patients and 7.3% (p-value = 0.0516) in healthy donors after *HBS1L* knockdown (compared with shNTC control) (Fig 5C, left, N = 3) on day 14. Notably, fetal hemoglobin levels were not significantly different in shNTC transduction compared to the untreated. The percentage of HbA has not changed under any circumstances (Fig 5C, middle). The level of HbA2/E was significantly decreased reciprocally (Fig 5C, right). Typical HPLC analysis of both healthy and patient were represented in Fig 5D and 5E.

The erythroid-specific transcriptional factors *BCL11A*, *GATA1*, *GATA2*, *KLF1*, *ZBTB7A*, *MYB*, and *ATF4* were then examined in untransduced, shNTC, and shRNA2 transduced erythroblasts (Fig 6). Also demonstrated was the quantity of *HBS1L* transcript that persisted in shNTC and shRNA2. The expression of *GATA1*, *GATA2*, *KLF1*, and *MYB* was relatively unchanged in all conditions. Notably, the levels of *ATF4* transcripts were remarkably increased, 3–4 fold, in both healthy donors and $\beta^0$-thalassemia/HbE patients. Both healthy and $\beta^0$-thalassemia/HbE erythroblasts transduced with shRNA2 showed a modest rise in *ZBTB7A*. The level of *BCL11A* transcript, on the other hand, slightly decreased as a result of transduction of shRNA2 only in $\beta^0$-thalassemia/HbE erythroblast.

## Discussion

There are 3 isoforms of HBS1L proteins, V1, V2, and V3, while V3 is the short isoform. Sankaran and colleagues [25] reported nonsense and splice site mutations of isoforms V1, V2 and V4, rendering null expression while preserving the expression of isoform V3 (S1 Fig). The authors further reported 2 patients with relatively normal hematological pictures as well as fetal hemoglobin level. These patients also exhibited various growth and skeletal abnormalities.

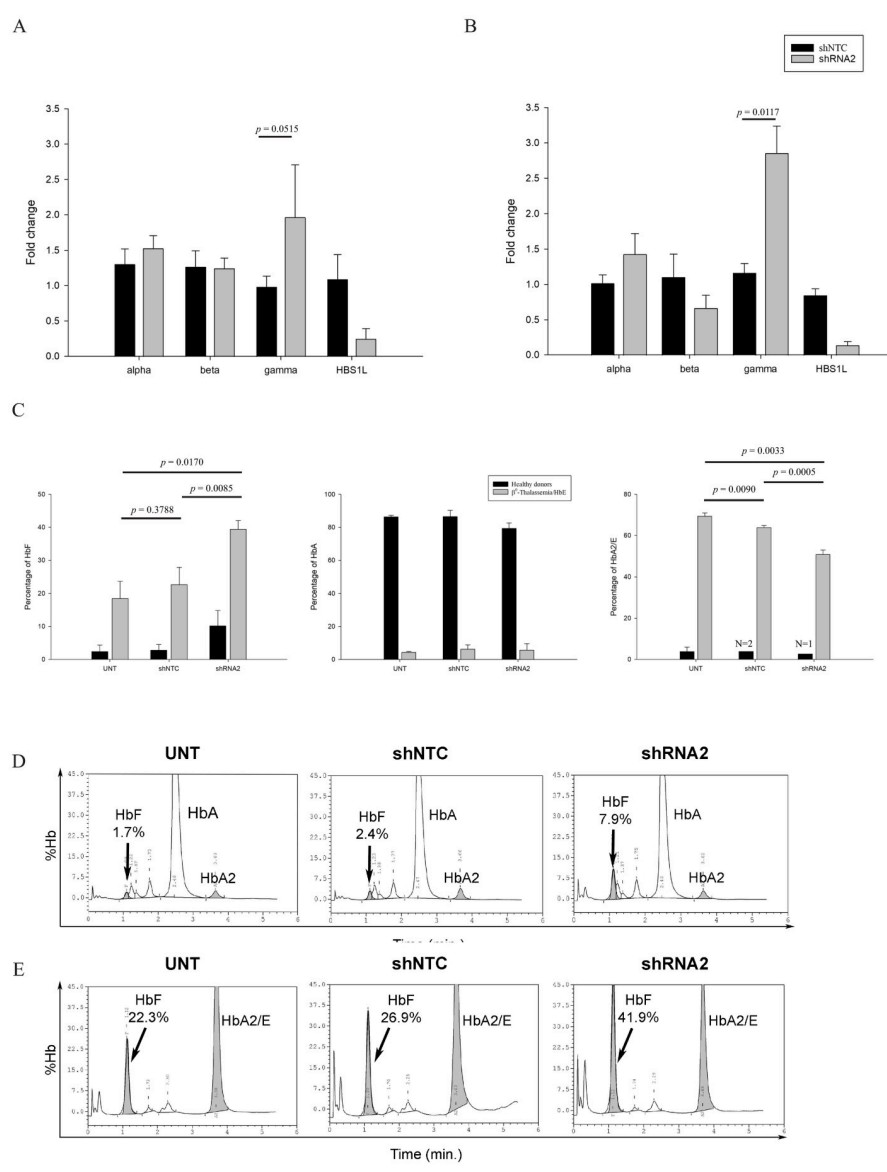

**Fig 5. Expression of α, β and γ-globin mRNAs after shRNA2 transduced erythroblast along with the amount of HBS1L remains.** Healthy donors on day 8, (A); β⁰-Thalassemia/HbE patients on day 10 (B) along with the amount of HBS1L remains. The percentages of HbF, HbA and HbA2/E analyzed by HPLC analysis on day 14 of erythroblasts from healthy donors and β⁰-thalassemia/HbE patients treated with shRNA2 (C). Data are represented as mean±SD. (N = 3, each). The representative chromatograms of HPLC analysis: healthy donor (D) and β⁰-thalassemia/HbE patient (E).

To our knowledge, this was the first study to demonstrate the expression of *HBS1L* during erythroid differentiation. Knocking down of all variants of HBS1L selectively upregulated mRNA levels 2–3 folds (when compared to the untransduced conditions) of γ-globin expressions in both β⁰-thalassemia/HbE and healthy groups. When compared to cells that had not been transduced, the knockdown of *BCL11A* was originally reported to result in a 5-fold activation of γ-globin per total β-like globin mRNAs [26].

A

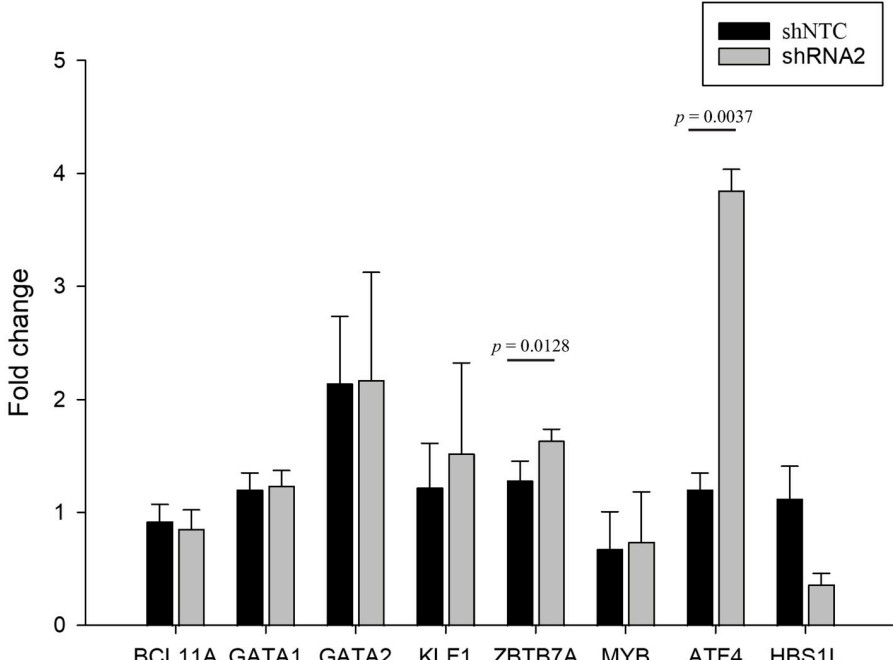

B

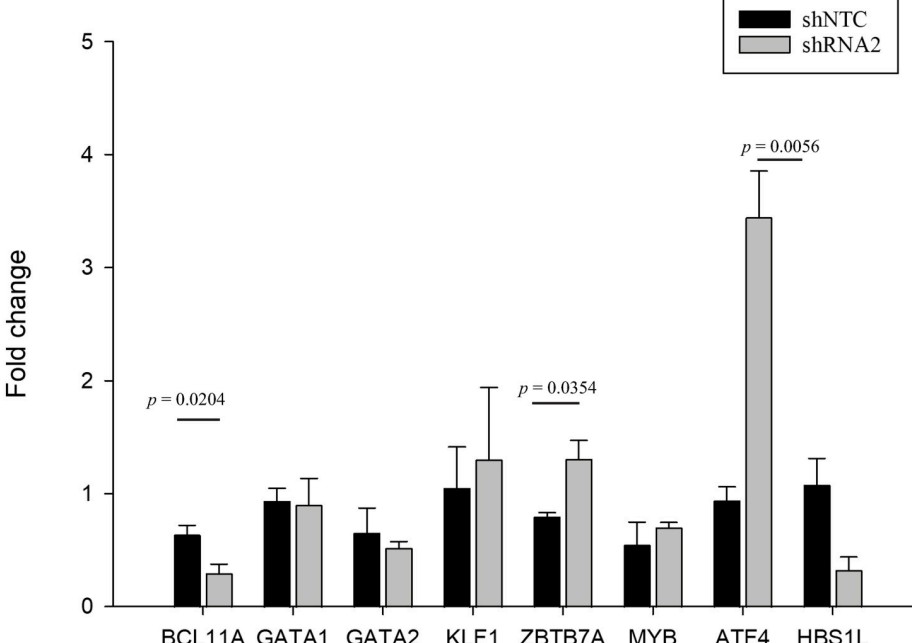

**Fig 6. Effect of *HBS1L* knockdown on selected erythroid related transcription factors in healthy donors (A) and β⁰-thalassemia/HbE patients (B).**

HBS1L together with PELO are specialized ribosome release factors involved in no-go and nonstop mRNA degradation. Endonucleolytic cleavage of the matching mRNA is initiated when the HBS1L/PELO complex binds a stalled ribosome [27].

ATF4 (Activating Transcription Factor 4) is a genuine direct target of NMD (nonsense mediated decay) since upregulation of its mRNA is caused by Rent1/Upf1 knockdown [28]. UPF1 serves as the focal point for bringing together other NMD factors and enlisting RNA decay factors [27]. In addition to the downstream reading frame that codes for the ATF4 protein, the *ATF4* mRNA possesses two additional upstream open reading frames (uORFs). Only uORFs are translated when there is no stress, which causes the ribosome to pause at a termination codon upstream of exon junction complex. This could explain why *ATF4* transcripts are vulnerable to NMD [28]. For Upf1 to mediate the connection with the RNA exosome complex and hence mediate NMD, Ski7 is required as an interacting factor in yeast [29]. Intriguingly, it has been discovered that the short form of HBS1L (HBS1LV3) serves as a human Ski7 homolog [30].

We suggest that HBS1LV3, like Ski7 in the Ski complex, associates with exosomes to boost UPF1 and cause the degradation of *ATF4* mRNA. According to our experiments, the abundance *ATF4* was increased after the knockdown of all HBS1L isoforms. As demonstrated by the induction of HbF by the protein phosphatase 1 inhibitor salubrinal in sickle erythroid progenitor cells, ATF4 is implicated in the elevation of γ-globin mRNA and an increase in HbF levels in erythroid progenitors via the eIF2α-ATF4 stress signaling pathway [31].

Studies utilizing the chromatin immunoprecipitation (ChIP) approach in erythroid precursors reveal a high GATA-1 binding signal in the middle of the intergenic *HBS1L-MYB* region on chromosome 6q23 [32]. It has been demonstrated that the presence of the rs66650371 minor allele in the intergenic region strongly correlates with *MYB* expression and reduced GATA-1 binding at this site in primary erythroid cells [33]. *HBS1L* expression, however, was not investigated. Additionally, RNAi knockdown of the transcription factors identified as residing in this region, LDB1, TAL1, and KLF1, suppresses *MYB* expression rather than *HBS1L* [32, 33].

We used lentiviral vectors carrying shRNA targeted at all major *HBS1L* transcript variants. The expressed shRNAs entered the RISC complex to cleave complementary mRNAs after being transformed into siRNA by Dicer. It was anticipated that the *GATA-1* transcript would continue to express itself. Indeed, we demonstrated that after *HBS1L* knockdown, the expression of *GATA-1* and *MYB* remained unaltered. The CRISPR-Cas9 method was also reported to induce Hb F without affecting *GATA-1* level when *ZNF410* was knocked down [34]. Together, our findings imply that GATA-1 and the *HBS1L-MYB* intergenic region might not be involved in the increase of HbF following *HBS1L* knockdown.

Reduction of *HBS1L* expression had a marginal increase in the percentages of basophilic erythroblasts assessed by flow cytometry. Our finding was attractive because most of fetal hemoglobin interventions so far have unfavorable effects on markedly delayed erythroid differentiation. The upregulation was also seen at the protein levels. Hemoglobin analysis showed that, on average, in healthy donors and patients with *HBS1L* knockdown, the percentages of fetal hemoglobin ranged from 2 and 18 to 10 and 40, respectively. Fetal hemoglobin levels prominently rose to 40% over basal when *BCL11A* was knocked down in healthy cells [26]. Alternatively, *HBS1L* knockdown provided a potential target for treating β-thalassemia as well as other hemoglobinopathies such as sickle cell disease.

## Supporting information

**S1 Fig. The diagram shows the three variants of the human HBS1L transcript.**
(TIF)

**S1 Table. Clinical data of the healthy donors and β⁰-thalassemia/HbE subjects recruited in this study.**
(DOCX)

**S2 Table. List of primer sequences used for qPCR in this study.**
(DOCX)

**S1 Raw images.**
(TIF)

## Acknowledgments

We appreciate the insightful criticism provided by three unnamed reviewers.

## Author Contributions

**Conceptualization:** Suthat Fucharoen, Chayanon Peerapittayamongkol.

**Data curation:** Chayanon Peerapittayamongkol.

**Funding acquisition:** Chayanon Peerapittayamongkol.

**Investigation:** Sukanya Chumchuen, Orapan Sripichai, Chayanon Peerapittayamongkol.

**Methodology:** Sukanya Chumchuen, Orapan Sripichai, Natee Jearawiriyapaisarn.

**Supervision:** Suthat Fucharoen, Chayanon Peerapittayamongkol.

**Validation:** Chayanon Peerapittayamongkol.

**Writing – original draft:** Sukanya Chumchuen, Chayanon Peerapittayamongkol.

**Writing – review & editing:** Sukanya Chumchuen, Orapan Sripichai, Natee Jearawiriyapaisarn, Suthat Fucharoen, Chayanon Peerapittayamongkol.

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
