## [Decision Letter · Decision Letter 0]

31 Mar 2022

PONE-D-21-35741Journal: PLOS ONEInduction of fetal hemoglobin: Lentiviral shRNA knockdown of HBS1L in β0-thalassemia/Hb E erythroid cells.PLOS ONE

Dear Dr. Peerapittayamongkol,

Thank you for submitting your manuscript to PLOS ONE. After careful consideration, we feel that it has merit but does not fully meet PLOS ONE’s publication criteria as it currently stands. Therefore, we invite you to submit a revised version of the manuscript that addresses all the points raised during the review process. I particularly recommend that authors provide a reasonable explanation for the discrepancy between their data and results previously reported by Sankaran et al; also additional experiments should  be performed according to relevant criticisms raised by both reviewers regarding the lack of an appropriate research design and controls. Please submit your revised manuscript by May 15 2022 11:59PM. If you will need more time than this to complete your revisions, please reply to this message or contact the journal office at plosone@plos.org. Please include the following items when submitting your revised manuscript:A rebuttal letter that responds to each point raised by the academic editor and reviewer(s). You should upload this letter as a separate file labeled 'Response to Reviewers'.A marked-up copy of your manuscript that highlights changes made to the original version. You should upload this as a separate file labeled 'Revised Manuscript with Track Changes'.An unmarked version of your revised paper without tracked changes. You should upload this as a separate file labeled 'Manuscript'.

We look forward to receiving your revised manuscript.

Kind regards,

Michela Grosso, Ph.D.

Academic Editor

PLOS ONE

Journal Requirements:

Reviewers' comments:

Reviewer's Responses to Questions

**Comments to the Author**

1. Is the manuscript technically sound, and do the data support the conclusions?

Reviewer #1: Partly

Reviewer #2: Partly

2. Has the statistical analysis been performed appropriately and rigorously? 

Reviewer #1: Yes

Reviewer #2: No

3. Have the authors made all data underlying the findings in their manuscript fully available?

Reviewer #1: Yes

Reviewer #2: Yes

4. Is the manuscript presented in an intelligible fashion and written in standard English?

Reviewer #1: Yes

Reviewer #2: Yes

5. Review Comments to the Author

Reviewer #1: In this paper, Peerapittayamongkol and colleagues build upon prior studies where they found correlation between a SNP in HBS1L and level of fetal hemoglobin (HbF) as well as disease severity in patients diagnosed with HbE β-thalassemia. As expression of HBS1L was reported to be reduced in culture differentiated erythroblasts from individuals with hereditary persistence of fetal hemoglobin (HPFH), the authors hypothesized that shRNA-mediated knockdown of HBS1L may lead to increased production of HbF in culture-differentiated erythroblasts of heathy volunteers and HbE β-thalassemia patients. That said, studies conducted on erythroblasts from HPFH individuals did not account for variations in genetic architecture of the globin locus or differential effects of known HbF repressor proteins. Also to the contrary, Sankaran and colleagues reported that complete loss of all major isoforms of HBS1L had no impact on hematological parameters or HbF levels (Sankaran et al, Blood (2013) 122 (23): 3845–384).

Here the authors, found that expression of HBS1L in culture-differentiated erythroblasts of heathy volunteers and HbE β-thalassemia patients was essentially indifferent, however, this experiment was only designed to assess total HBS1L rather than discriminate between total and alternative splice variants of this gene product (V1, V2, V3). This is particularly important as the authors provide an argument for significance of the V3 isoform in the discussion as a contradiction to prior published findings (Sankaran et al, Blood (2013) 122 (23): 3845–384). They went on to show that shRNA knockdown of HBS1L increased levels of gamma-globin mRNA and HbF in differentiated erythroblasts of heathy volunteers and HbE β-thalassemia patients without significant impact on erythroid differentiation or cell morphology. That said, they did not evaluate the expression of fetal hemoglobin per cell by flow cytometry, ability of these cultured cells to reach full maturity as demonstrated by enucleation, or compare total hemoglobin for HBS1L KD β-thalassemia cells versus healthy donors. Finally, the authors do not expand upon these initial finds by delving into the potential responsible mechanism(s) to include changes in expression of: 1) known HbF repressors or activators, 2) chromatin modifiers, or 3) erythroid-specific transcription factors that may be affected by loss of a protein involved in mRNA stability, which limits the overall significance of these findings.

Major comments

lines 65 – 68: The description of HBS1L function is difficult to appreciate from the provided text. Please revise these two sentences to clarify this information.

Supplementary Table 1. It is preferable to use “healthy donor” as an alternative to “normal”. Patients in this study are also “normal” they just have thalassemia. Please replace N = normal with HD = Healthy donor throughout the manuscript to include figures and figure legends.

lines 116 – 141. The description of the shRNAs, construction of lentiviral vector, and production and transduction can be combined into a single section.

lines 148 – 162. The description of RNA isolation, reverse transcription and qPCR can be combined into a single section.

Figures 1 and 2. Please combine these data into a single figure where panel A demonstrates results for the 3 healthy donors and panel B for the 4 thalassemia patients. It would be beneficial if the scale of the Y-axis were the same for panels A and B so that differences in mRNA expression can be easily appreciated by the reader. This figure also needs a descriptive legend summarizing the information presented in each panel. Please replace N = normal with HD = Healthy donor for the reason described above. Finally, it would be beneficial to place into context the mean expression level ± S.D. for the healthy donors vs. patients at each time point. Were any of the differences in expression statistically significant?

Figures 3 and 4. Please combine these data into a single figure where panel A demonstrates results for HBS1L mRNA level and panel B protein level. This figure also needs a descriptive legend summarizing the information presented in each panel and could benefit from a schematic diagram shown the timing of lentiviral transduction, cell expansion and differentiation as well as collection of samples for isolation of mRNA and protein. It is odd that levels of HBS1L mRNA were unchanged relative to untreated cells by qPCR but protein levels appear to be reduced in both the cytoplasmic and nuclear fractions even though equivalent amounts of protein appear to be loaded as demonstrated by lamin and actin. Do the authors have any explanation for this result?

Figures 5 and 6. Please combine these data into a single figure where panel A - D demonstrates flow cytometry results and panels E and F photomicrographs of cell cytospins. This figure also needs a descriptive legend summarizing the information presented in each panel.

Figures 7, 8, and 9. Please combine these data into a single figure where panel A and B demonstrates results for alpha, beta and gamma globin expression for the healthy donors and thalassemia patients, respectively. It would be beneficial if the scale of the Y-axis were the same for panels A and B so that differences in expression can be easily appreciated by the reader. Figure 8 can be presented as panel C with results plotted as bars demonstrating mean ± SD for each condition as an alternative to the line graph which is somewhat confusing. Finally, HPLC results can be presented as panels D and E, respectively. This figure also needs a descriptive legend summarizing the information presented in each panel.

Minor comments

line 29: remove space in “Hb E” to read “HbE” and remove the word “erythroid”

line 30: replace “on” with “of”

line 39: Change “Asea” to “Asia”

line 45: remove space in “Hb E” to read “HbE”

line 47: citation (6) appears to have a reduced font size

line 47: remove space in “Hb F” to read “HbF”

line 49: replace “transplantation” with “transplant”

line 54: replace “amending” with “amend”

lines 45-57: combine text into a single paragraph

line 59: remove the word “in” and replace “reveals” with “revealed”

line 61: remove space in “Hb E” to read “HbE”

line 62: remove “the” and remove space in “Hb F” to read “HbF”

line 64: remove “the” and remove space in “Hb F” to read “HbF”

line 70: remove space in “Hb E” to read “HbE”

line 75: remove space in “Hb F” to read “HbF”

line 84: remove the word “normal”

line 85: remove space in “Hb E” to read “HbE”

line 86: replace “normal” with “volunteers” or “subjects”

line 90: “Mmononuclear” should read “Mononuclear”

line 121: % appears to have a reduced font size

line 282: replace “expressions” with “expression”

lines 283 and 286: replace “normal” with “healthy controls” or “healthy subjects”

line 302: replace “normal” with “healthy”

Reviewer #2: Hemoglobin disorders including beta-thalassemia can be cured by hematopoietic stem cell (HSC) gene therapy with beta-globin gene addition or short hairpin RNA (shRNA)-based fetal hemoglobin (HbF) induction. BCL11A gene, LRF gene, HBS1L-MYB region, and other HPFH (hereditary persistence of fetal hemoglobin) mutations are main targets for HbF induction. In this study, the authors demonstrated that shRNA-based knockdown of the HBS1L gene can induce HbF expression in CD34+ cell-derived erythroid cells in vitro (from healthy donors and beta0-thalassemia / hemoglobin E patients); however, the HbF induction levels were relatively low at 5-10% in healthy donor cells (1% at the baseline) and 35-40% in patients’ cells (20% at the baseline).

Major points

1. The authors should demonstrate the shRNA-target sites (shRNA1 and shRNA2) on a schema of the HBS1L gene and RNA variants. In addition, the authors should demonstrate the target sequences of shRNA1 and shRNA2.

2. The authors should compare HbF amounts at the protein level in CD34+ cell-derived erythroid cells between HBS1L knockdown and BCL11A knockdown.

3. The authors should evaluate erythroid-specific transcriptional factors (such as GATA1, GATA2, KLF1, BCL11A, and MYB) in erythroid cells with or without knockdown of HBS1L.

4. The authors should demonstrate a schema of methods regarding cell type, conditions of lentiviral transduction, and methods of erythroid differentiation for Figures 1-2, Figures 3-4, Figures 5-6, and Figures 7-9.

5. In Figure 1-2, the authors should demonstrate whether erythroid cells are sufficiently matured for hemoglobin analysis, such as cell surface analysis for CD71 and CD235a.

6. In Figure 3, the authors should demonstrate HBS1L RNA amounts in un-transduced cells.

7. In Figure 4, the authors should explain why HBS1L protein amounts were reduced in the shNTC control.

8. In Figures 3-4, Figures 5-6, and Figures 7-9, the authors should demonstrate transduction efficiency (such as vector copy number) in transduced cells.

9. In Figure 5, the author should demonstrate vector copy number (or HbF induction) among each fraction, if lentiviral transduction efficiency in bulk cells is not ~100%.

10. In Figure 6, the authors should measure the percentage of enucleation among groups.

11. In Figure 7, the authors should demonstrate gamma-globin RNA amounts in un-transduced cells.

12. In Figure 8, the authors should demonstrate a graph of the average and standard deviation with statistical analysis regarding the absolute amounts (percentages per total hemoglobin) of HbF, HbA, and HbA2/E.

Minor pints

1. In Methods, the authors should define the pLL3.7 lentiviral vector.

2. In Methods, the authors should explain the more detailed methods for HPLC.

3. The authors should combine Figures 1 and 2, Figures 3 and 4, Figures 5 and 6, as well as Figures 7, 8, and 9.

4. In the Figure 3 legend, the author should explain how to calculate HBS1L RNA amounts.

6. PLOS authors have the option to publish the peer review history of their article (what does this mean?). If published, this will include your full peer review and any attached files.

Reviewer #1: No

Reviewer #2: No

---

## [Author Response · Author response to Decision Letter 0]

27 Aug 2022

We sincerely appreciate the Editor's, Reviewer 1's, and Reviewer 2's feedback and ideas.

---

## [Decision Letter · Decision Letter 1]

16 Nov 2022

PONE-D-21-35741R1Induction of fetal hemoglobin: Lentiviral shRNA knockdown of HBS1L in β0-thalassemia/HbE erythroid cells.PLOS ONE

Dear Dr. Peerapittayamongkol,

Thank you for submitting your manuscript to PLOS ONE. After careful consideration, we feel that it has merit but does not fully meet PLOS ONE’s publication criteria as it currently stands. Therefore, we invite you to submit a revised version of the manuscript that addresses the points raised during the review process.

The authors have properly addressed the comments raised during the first round of revision and have substantially improved the overall quality of the manuscript.

However, before being considered for acceptance, the manuscript should be revised (minor revision) in order to address the comment raised by Reviewer 3: "Interactions of the erythroid-specific transcription factor GATA-1 have been demonstrated with several sites in the HBS1L-MYB interval. Since the authors mention that the expression of GATA-1 was relatively unchanged, this has to be incorporated and explained in the discussion."

We look forward to receiving your revised manuscript.

Kind regards,

Michela Grosso, Ph.D.

Academic Editor

PLOS ONE

Journal Requirements:

Additional Editor Comments:

The authors have properly addressed the comments raised during the first round of revision and have substantially improved the overall quality of the manuscript.

However, before being considered for acceptance, the manuscript should be revised (minor revision) in order to address the comment raised by Reviewer 3: "Interactions of the erythroid-specific transcription factor GATA-1 have been demonstrated with several sites in the HBS1L-MYB interval. Since the authors mention that the expression of GATA-1 was relatively unchanged, this has to be incorporated and explained in the discussion."

Reviewers' comments:

Reviewer's Responses to Questions

**Comments to the Author**

1. If the authors have adequately addressed your comments raised in a previous round of review and you feel that this manuscript is now acceptable for publication, you may indicate that here to bypass the “Comments to the Author” section, enter your conflict of interest statement in the “Confidential to Editor” section, and submit your "Accept" recommendation.

Reviewer #3: All comments have been addressed

2. Is the manuscript technically sound, and do the data support the conclusions?

Reviewer #3: Yes

3. Has the statistical analysis been performed appropriately and rigorously? 

Reviewer #3: Yes

4. Have the authors made all data underlying the findings in their manuscript fully available?

Reviewer #3: Yes

5. Is the manuscript presented in an intelligible fashion and written in standard English?

Reviewer #3: Yes

6. Review Comments to the Author

Reviewer #3: It is well designed and writen paper that presents the ability of shRNA-induced HBS1L knockdown to increase HbF in thalassemic erythroblasts and furthermore to slightly improve erythroid differentiation.

One suggestion: Interactions of the erythroid-specific transcription factor GATA-1 have been demonstrated with several sites in the HBS1L-MYB interval. Since the authors mention that the expression of GATA-1 was relatively unchanged, this has to be incorporated and explained in the discussion.

7. PLOS authors have the option to publish the peer review history of their article (what does this mean?). If published, this will include your full peer review and any attached files.

Reviewer #3: No

---

## [Author Response · Author response to Decision Letter 1]

6 Dec 2022

I want to thank Reviewers 1 and 2 for addressing the important issues in the earlier submission. I sincerely appreciate Reviewer 3's remarks on our reviewer response and the interest to discuss about the relationship between GATA-1 and the HBS1l-MYB intergenic region. Additional discussion has been added to line number 375-390 of the present revision.

---

## [Decision Letter · Decision Letter 2]

17 Jan 2023

Induction of fetal hemoglobin: Lentiviral shRNA knockdown of HBS1L in β0-thalassemia/HbE erythroid cells.

PONE-D-21-35741R2

Dear Dr. Peerapittayamongkol,

We’re pleased to inform you that your manuscript has been judged scientifically suitable for publication and will be formally accepted for publication once it meets all outstanding technical requirements.

Kind regards,

Michela Grosso, Ph.D.

Academic Editor

PLOS ONE

Additional Editor Comments (optional):

Reviewers' comments:

Reviewer's Responses to Questions

**Comments to the Author**

1. If the authors have adequately addressed your comments raised in a previous round of review and you feel that this manuscript is now acceptable for publication, you may indicate that here to bypass the “Comments to the Author” section, enter your conflict of interest statement in the “Confidential to Editor” section, and submit your "Accept" recommendation.

Reviewer #3: All comments have been addressed

2. Is the manuscript technically sound, and do the data support the conclusions?

Reviewer #3: Yes

3. Has the statistical analysis been performed appropriately and rigorously? 

Reviewer #3: Yes

4. Have the authors made all data underlying the findings in their manuscript fully available?

Reviewer #3: Yes

5. Is the manuscript presented in an intelligible fashion and written in standard English?

Reviewer #3: Yes

6. Review Comments to the Author

Reviewer #3: The authors have adequately addressed the comments and i think that the manuscript is now acceptable for publication

7. PLOS authors have the option to publish the peer review history of their article (what does this mean?). If published, this will include your full peer review and any attached files.

Reviewer #3: No

---

## [Editor Report · Acceptance letter]

27 Feb 2023

PONE-D-21-35741R2 

Induction of fetal hemoglobin: Lentiviral shRNA knockdown of *HBS1L* in β^0^-thalassemia/HbE erythroid cells. 

Dear Dr. Peerapittayamongkol:

I'm pleased to inform you that your manuscript has been deemed suitable for publication in PLOS ONE. Congratulations! Your manuscript is now with our production department. 

Kind regards, 

on behalf of

Prof. Michela Grosso 

Academic Editor

PLOS ONE